# Selection of Conditions in PVB Polymer Dissolution Process for Laminated Glass Recycling Applications

**DOI:** 10.3390/polym14235119

**Published:** 2022-11-24

**Authors:** Marek Królikowski, Piotr Żach, Maciej Kalestyński

**Affiliations:** 1Faculty of Chemistry, Warsaw University of Technology, Noakowskiego 3, 00-664 Warsaw, Poland; 2Faculty of Automotive and Construction Machinery, Warsaw University of Technology, Narbutta 84, 02-524 Warsaw, Poland; 3Wasatech Recycling sp. z o. o., Krakowiaków 68-70, 02-255 Warsaw, Poland

**Keywords:** polyvinyl(butyral), recycling PVB, waste PVB, laminated glass, solution of PVB

## Abstract

Polyvinyl(butyral) (PVB) post-production waste collected from the windshields of end-of-life vehicles and post-consumer building laminated glass are valuable polymeric materials that can be reused. Every year, large amounts of PVB waste are still being buried in landfills owing to a lack of appropriate recycling techniques. Before reuse, PVB should be thoroughly cleaned of solid contaminants such as glass dust, fused heating wires, and other waste polymers, metals, and ceramics. This can be done by polymer dissolution and filtration. In this study, we propose the purification of PVB from contamination by dissolving the post-consumer polymeric materials into single and binary organic solvents. As part of the experimental work, measurements and optimization of the dissolution time of PVB were performed. PVB dissolves faster when a binary solvent (2-propanol + ethyl acetate) than pure 2-propanol is used. From the point of view of the practical application of PVB solutions, measurements of density and dynamic viscosity as a function of PVB concentration and temperature were performed. The PVB solutions obtained in this work can be widely used as glues for glass, ceramics, metal, impregnating, and insulating materials or as paint additives that are entirely transparent for visible light and to block UV rays.

## 1. Introduction

Environmental aspects, increasing growth of virgin polymer prices, and still growing landfill fees bring about the increasing interest encountered in plastic waste recycling [1]. The problem has been solved at a global level for common plastics such as polyethylene, polypropylene, poly(ethylene terephthalate), and poly(styrene) [2,3,4]. However, poly(vinyl butyral) PVB does not belong to this group, although it is commonly used in the glass lamination process, especially as an interlayer in car windshields, architecture (facades and construction), and security glass [5,6,7,8]. PVB is also used as an interlayer in the construction and encapsulation of solar cells [9,10,11]. PVB is a polymer material with excellent properties, including excellent optical clarity, high adhesion strength to glass, metals, and ceramics, high mechanical strength, and high deformation. PVB for laminated glass is currently manufactured and commercialized by companies worldwide, among which should be mentioned: Eastman Chemical Company (US), Kuraray (Germany), Sekisui Chemicals (Japan), Everlam (Belgium), Genau Manufacturing Company (India), KB PVB (China), Chang Chun Group (China), DuLite (China), and Tiantai Kanglai Industrial (China) [12].

Currently, the major source of waste PVB is the windshields of end-of-life vehicles and also side windows because, by 2018, a third of car models had laminated side windows instead of tempered windows. The total number of end-of-life vehicles reported in the European Union reached 6.1 million in 2018, sharply rising from the 4.8 million recorded in 2016 and 5.3 million in the year 2017, and was the highest since 2010 [13]. This shows that more and more waste PVB will be collected in landfills each year. Most countries limit the recycling of automotive windshields, architecture windows, and solar panels to aluminum and glass, and PVB interlayer is a less important waste.

Current researchers and recycling companies are focused on the recycling of interlayer PVB of laminated glass, mainly from the automotive sector [14,15,16,17]. To the best of our knowledge, so far, only two manufacturers offer 100% recycled PVB films or powder [18,19]. The interlayers are manufactured from collected postindustrial PVB trimmings. By carefully sorting and reprocessing PVB trimmings generated during the production of laminated glass, the Butacite^®^ G sheeting is a clean, reliable raw material for use with new safety glass laminates. 

Miloš Matúš et al., in publication [15] and patent [20], proposed an economically efficient process. The decomposition of laminated glass is based on the principle of retaining the integrity of the PVB film and is carried out on the process line made up of modules: the breaking module in which the glass is broken in both transverse and longitudinal directions, the vibration module in which a broken but compact car windshield can be shaken, and the stripper module in which mechanical cleaning of the PVB film is carried out. At the end of the process line, an additional washing module can be used. This is a typical approach where the PVB is mechanically separated from glass, and the final product is film or flakes of PVB. It is probably the best solution to recycle PVB, but this would not remove the impurities embedded in it, such as resistance wire, which is used to heat the windshield. Another module in a process line should be added, where the PVB can be dissolved and the solid contaminants filtered out.

The PVB polymer chain has both hydrophobic groups (butyral or acetyl group) and hydrophilic groups (hydroxyl residues). This composition allows the PVB to dissolve in polar and a mixture of polar and nonpolar organic solvents. PVB should be soluble or partially soluble in the following solvents: acetic acid, acetone, methanol, ethanol, 2-propanol, butanol, 2-butoxyethanol, cyclohexanone, benzyl alcohol, 1-methoxy-propanol-2, butyl glycol, n-butyl acetate, ethyl acetate, N,N-dimethylacetamide (DMA), N,N-dimethylformamide (DMF), N,N-dimethylsulfoxide (DMSO), N-methylpyrrolidone (NMP), and tetrahydrofurane (THF), but solubility is strongly dependent on the average molecular mass, composition and the vinyl acetate content [21]. In the literature, various solvents or a mixture of binary solvents to dissolve the PVB were used. Daniela Lubasova and Lenka Martinova dissolved PVB in different types and quantities of solvents and their mixtures (methanol, ethanol, THF, DMSO) [22]. Wenwen Luan et al. measured the intrinsic viscosities of PVB in ethanol/water solutions in the temperature range of 293.15–313.15 K [23]. Ryoichi Furushima et al. used a mixed organic solvent consisting of xylene and 2-propanol [24]. However, some of these solvents are not environmentally friendly; for example, THF, NMP, and xylenes are toxic, as is methanol. In the literature, the dissolution time in the proposed solvents, the effect of temperature on dissolution, and physicochemical parameters: density, dynamic viscosity, and refractive index of the PVB solutions are not presented.

In this work, the PVB polymer into the pure solvent, 2-propanol, and binary solvent mixture {{2-propanol + ethyl acetate, *V*/*V* = 1/1} were dissolved. In the solubility measurements, two sizes of polymer flakes: 30.0 × 30.0 × 0.82 mm and 10.0 × 10.0 × 0.82 mm, were assumed. Similarly, sized polymer flakes are obtained in the processes of mechanical shredding of car windshields and laminated glass. As part of the experimental work, measurements and optimization of the dissolution time of PVB were performed. From the point of view of the practical application of PVB solutions, measurements of density and dynamic viscosity as a function of PVB concentration and temperature were performed. 

The prepared PVB-based solutions can be used as impregnates, varnishes, and glue for glass, metal, wood, paper, leather, cloth, and other materials. The reuse of PVB, which is currently landfilled, is of environmental importance. It allows for the management of waste material, the storage of which currently has to be paid for. The costs for the preparation of solutions are as follows: solvent, IPA, and ethyl acetate amounting to 1300 €/t [25], the energy of purifying PVB from the glass, and the costs of the dissolution process: heating the solvent, stirring, or ultrasound. 

## 2. Materials and Methods

The specific information on all the chemicals used in the experiment is listed in Table 1. The commercially available PVB with ${\bar{M}}_{w}=140,000\;g/\mathrm{mol}$ was obtained from Pilkington Automotive (Poland), where the polymer is used for laminated safety car glass. The image of the testing sample is presented in Appendix A in the Appendix A (Appendix A). The structure of PVB was confirmed by FT-IR analyses. All the solvents were used without further purification, and the purity was checked by gas chromatography (GC). The purity of the solvents was not lower than that declared by the supplier.

FT−IR Spectrometry and UV-VIS Spectroscopy

The Fourier transform-infrared spectrum analyses of the polymer were carried out using Nicolet iS5, a Thermo Scientific Mid Infrared FT−IR spectrometer equipped with iD7 ATR Optical Base. After washing the ATR crystal with ethanol, the samples of PVB were directly placed on the surface of the ATR. The wavenumber ranged from 3900 to 400 cm^−1^. The solution of PVB was characterized by UV−VIS spectroscopy using Genesys 180, Thermo Scientific ranging from 190 to 1100 nm. 

Differential Scanning Calorimetry

Glass transition temperature (*T_g_*) and heat capacity change at the glass transition temperature (Δ*C_p_*_(*g*)_) have been determined using differential scanning calorimetry, DSC technique. The experiments were performed with DSC 1 STARe System (Mettler Toledo) calorimeter equipped with a liquid nitrogen cooling system and operating in a heat-flux mode. The sample cell was constantly fluxed with high purity nitrogen at a constant flow rate of 20 mL·min^−1^. The apparatus was calibrated with the 99.9999 mol% purity indium sample and with high purity ethylbenzene, *n*-octane, *n*-decane, *n*-octadecane, *n*-eicosane, cyclohexane, biphenyl, and water. The calibration experiment was carried out with a 5 K·min^−1^ heating rate in the temperature range from 180 to 450 K. The sample was sealed in hermetic aluminum pans. An empty pan was used as a reference. Measurements were taken three times, each time for a new PVB sample. The average sample mass was 9.9 mg throughout this study. The experiments were performed at a heating rate of 5 K∙min^−1^. Measurements were performed and ranged from 240 to 360 K.

Solubility of PVB

For the measurement of PVB solubility, polymer samples with dimensions of 30.0 (±0.1) × 30.0 (±0.1) mm or 10.0 (±0.1) × 10.0 (±0.1) mm and a thickness of 0.82 (±0.02) mm were prepared. Each of the samples was weighed on AB204-S, Mettler Toledo analytical balance with an uncertainty of 1 × 10^−4^ g. 10, 15, and 30 mL of pure 2-propanol or binary solvent mixtures {2-propanol + ethyl acetate, *V*/*V* = 1/1} were used to dissolve PVB. Knowing the PVB mass, volume, and density of the solvent, the final solution concentration could be calculated. PVB dissolution processes were prepared in a jacketed and thermostatted glass vessel with a volume of 50 mL. The jackets were connected to the thermostatic water bath, Julabo CORIO CD-BC6, to maintain a constant temperature with an accuracy of *T* = 0.05 K. The heterogeneous mixtures of dissolving PVB and solvent were vigorously stirred with a coated magnetic bar. Magnetic stirrer IKA RCT basic was used. The rotation speed of the stirrer was constant and equal to 500 rpm, which was enough to provide perfect contact between the PVB and the solvent. In the next part of the experiment, an ultrasonic bath, PROCLEAN 2.0M ECO with a capacity of 2 dm^3^, and ultrasound power of 60 W were used instead of a mechanical stirrer. Samples of the solution were taken at specified intervals using 1 mL syringes with a steel needle. The refractive index of the PVB solution was measured. The concentration of PVB was determined using the calibration curve of the refractive index. Solubility measurements were performed in the range from 0 to 20%*wt* PVB. Above this range, the polymer flakes do not dissolve, and a gel-like swollen layer on PVB is formed.

Density

The temperature-dependent density of liquid PVB solution was determined under ambient pressure using a vibrating tube densimeter—DMA 4500 M, Anton Paar. The densimeter has an automatic correction of the viscosity of the liquid sample. Doubly distilled and degassed water and dried air was used for calibration. Two integrated Pt 100 platinum thermometers allow for precision temperature control within 0.05 K. Density was measured with resolution 1·10^−5^ g∙cm^−3^, but considering the impurities of the samples and solvents, the measurement uncertainty was estimated to be better than 5·10^−4^ g∙cm^−3^. The density of PVB solutions was measured in the range from 0.7494 to 0.8542 g∙cm^−3^.

Dynamic viscosity

The dynamic viscosity of dissolved PVB was determined using an AMVn, Anton Paar rheometer based on the “falling ball” principle. Before the experiment, a liquid standard provided by the supplier was used for the apparatus calibration. The temperature was controlled with a precision of 0.05 K. The relative standard uncertainty of the measured viscosity was estimated to be 5%. Over the measured viscosity range, capillaries of varying diameters were used depending on the viscosities of the fluid. The diameter of the capillary was 1.6 mm (ball diameter 1.5 mm) for viscosity in the range from 0.3 to 10 mPa·s, 1.8 mm (ball diameter 1.5 mm) for viscosity in the range from 2.5 to 70 mPa·s, and 3.0 mm (ball diameter 2.5 mm) for viscosity in the range from 20 to 230 mPa·s, and 4.0 mm (ball diameter 2.5 mm) for viscosity in the range from 200 to 2500 mPa·s. The dynamic viscosity of PVB solutions was measured in the range from 0.639 to 940 mPa·s.

Refractive index

The refractive index of the liquid PVB solution was determined by a precision Carl Zeiss Abbe Refractometer Type G with an accuracy of 5·10^−5^ at *T* = 298.15 ± 0.05 K. A calibration curve was made for each mixture, and the accuracy of the composition determination was better than 0.05%*wt* PVB. The refractive index of PVB solutions was measured in the range from 1.37083 to 1.38521.

## 3. Data Modeling

The experimental data of PVB dissolution in a pure or binary mixture of solvents were fitted to a quadratic equation:(1)$$\%wt=A_{1}\cdot t^{2}+A_{2}\cdot t$$
where %*wt* and *t* (h) are the PVB weight percent in a solvent and time, respectively. The adjustable parameters of the equation, *A*_1_ and *A*_2,_ along with the average absolute relative deviation (*AARD*), are listed in Appendix A in Appendix A. The *AARD* is defined as:(2)$$AARD\left(\%\mathrm{wt}\right)=\frac{1}{n}\overset{n}{\underset{i=1}{\sum }}\left|\frac{\%wt^{exp}-\%wt^{cal}}{\%wt^{exp}}\right|$$
where $\%wt^{exp}$ and $\%wt^{cal}$ are the experimental and calculated values of PVB weight percent and *n* is the total number of data points.

The concentration and temperature dependence of density, *d*, and dynamic viscosity, *η*, for each system under study was described using the following equations [26]:(3)$$d/\left(g\cdot cm^{-3}\right)=B_{1}\cdot \%wt+B_{2}$$
where parameters ***B*_1_** and ***B*_2_** are linear functions of temperature:(4)$$B_{1}=b_{11}\cdot T/K+b_{12}$$
(5)$$B_{2}=b_{21}\cdot T/K+b_{22}$$
where $b_{11},b_{12},b_{21}$, and $b_{22}$ are regressed parameters.
(6)$$\eta /\left(mPa\cdot s\right)=\exp\left(C_{1}+\frac{C_{2}}{T/K}\right)$$
where parameters $C_{1}$ and $C_{2}\;$ are linear functions of concentration:(7)$$C_{1}=c_{11}\cdot \%wt+c_{12}$$
(8)$$C_{2}=c_{21}\cdot \%wt+c_{22}$$
where $c_{11},c_{12},c_{21}$ and $c_{22}$ are calculated parameters. A simple linear equation has been successfully used to describe the refractive index:(9)$$n_{D}=D_{1}\cdot \%wt+D_{2}$$

The quality of the data fit was determined using the root mean square error (*RMSE*):(10)$$\sigma \left(X\right)=\sqrt{\frac{\sum_{i=1}^{N}{\left(X^{exp}-X^{cal}\right)}^{2}}{N-k}}$$
where $X^{exp}$ and $X^{cal}$ are the experimental and calculated values of a density, dynamic viscosity, and refractive index. *N* and *k* are the total numbers of data points and parameters, respectively. The adjustable parameters with *RMSE* are presented in Appendix A in Appendix A. The useful uses of the correlations are to provide an expression for interpolating the experimental information.

## 4. Results and Discussion

### 4.1. Analysis of PVB

The structure of PVB was confirmed by infrared spectroscopy. The FT−IR spectrum of PVB is presented in Figure 1. The PVB molecule contains a CH_2_ backbone and different functional groups, which all have their typical vibrations. The IR bands at 2957, 2935, and 2870 cm^−1^ were the valence vibrations of CH_2_-groups of the polymer backbone. Corresponding deformation modes appeared in 1458, 1434, and 1380 cm^−1^. Free hydroxyl groups showed characteristic broadband between 3500 and 3200 cm^−1^ as well as the deformation mode at 1342 cm^−1^. A distinctive peak of the spectra appeared at 1731 cm^−1^. This band and also the peak at 1240 cm^−1^ are caused by the carbonyl group of ester. The strong vibration between 1200 and 1000 cm^−1^ is derived from the acetal groups. This FT−IR spectrum is consistent with the literature spectra [9,27].

The DSC thermogram of the analyzed PVB is presented in Appendix A in the Appendix A. As reported in the literature, un−plasticized PVB had a glass transition temperature, *T*_g_, as measured by DSC and corrected for a thermal lag of 347 ± 2 K [28]. All commercial PVB used in glass lamination exhibited *T*_g_ = 289 ± 2 K. The addition of 25%*wt* plasticizers, such as dibutyl sebacate and butyl phenyl phthalate, lowered the glass transition temperature. The PVB tested in this work showed the glass transition temperature, *T*_g_ = 289.8 K, and the heat capacity change at glass transition, $\Delta C_{p(g\}}=0.39\;J\cdot g^{-1}\cdot K^{-1}$. The *T*_g_ is consistent with the PVB used in automotive windshields and reported in the literature [29].

### 4.2. Dissolution of PVB

The dissolution of a PVB in a pure solvent or binary mixture of solvents involves two transport processes, namely solvent diffusion and chain disentanglement [30]. The solvent will be diffused into an uncrosslinked, amorphous, glassy PVB if the polymer is in contact with the thermodynamically compatible solvents. Due to the plasticization of the polymer by the solvent, a gel-like swollen layer is formed along with two separate interfaces, one between the glassy polymer and gel layer and the other between the gel layer and the solvent. After some time, an induction time, the polymer dissolves. In this study, the dissolution time of the polymer PVB was determined. In the first step, the temperature, *T* = 298.15 K, the stirrer speed, 500 rpm, and the size of the polymer flakes: 30.0 × 30.0 × 0.82 m, were assumed to be constant. Three different initial solvent volumes were tested: 10, 15, and 30 mL obtaining different final concentrations of PVB solution.

The experimental results are listed in Appendix A in the Appendix A, and the graphical presentation is in Figure 2. The total polymer dissolution time under these conditions is *t* = 10 h. Different initial solvent volumes do not affect the dynamics and time of complete dissolution of the PVB polymer. From Figure 2b, it can be seen that after 2 h, 35%*wt* of PVB was dissolved in the case of the initial amount of solvent 10 mL and 15 mL, and about 23%*wt* of PVB for 30 mL. After 4 h of dissolution, it is 66%*wt*, 62%*wt*, and 57%*wt* of PVB for 15 mL, 10 mL, and 30 mL of the initial amount of solvent, respectively. After the next 4 h, it is 97%*wt*, 96%*wt*, and 91%*wt* of PVB for 15 mL, 10 mL, and 30 mL of the initial amount of solvent, respectively. The results clearly show that the rate of PVB dissolution with the passage of time decreases. It is assumed that at the first step, the solvent flux into the polymer is sufficient to carry away all the chains that dissolved from the gel into the liquid. The dissolution is disassociation-controlled. When the solvent concentration at the gel–liquid boundary reaches a constant concentration, the flux is no longer sufficient to carry all of the dissolved chains away from the surface, at which time the dissolution becomes diffusion-controlled [30]. However, the dissolving PVB is observed only in the range from 0 to 20%*wt*. Above this range, the viscosity of the solution sharply increases, and the polymer flakes do not dissolve, while only a gel-like swollen layer on PVB is formed. 

A much more significant influence on the dynamics and time of complete dissolution of PVB in 2-propanol has the size of the PVB flakes. Reducing the fraction from 30.0 × 30.0 × 0.82 mm to 10.0 × 10.0 × 0.82 mm reduces the complete dissolution time to 6.5 h under the same conditions, as shown in Figure 3 and in Appendix A in Appendix A.

This is mainly due to increased polymer-solvent contact area and easier plasticization of the polymer by the solvent. Devotta et al. investigated the relation of the polymer size to various parameters in polystyren/cyclohexane and polymethyl methacrylate/benzene systems. The dissolution time was almost constant for particles ranging from a few microns to 50 μm and increased for larger particle sizes [31,32]. It follows that the polymer should be further shredded and cut to achieve smaller particles of PVB and shorter dissolution times. However, obtaining such a small polymer fraction is not easy. The process requires shredding mills with sharp knives and cooling. PVB is soft, flexible, and sticks above the glass transition temperature, *T_g_* = 289 K. Therefore, it must be cut at a low temperature using cryogenic apparatus. The knives of shredding mills can be exterminated because glass dust and heating wires in the recycled material occurred. Therefore, obtaining a dust fraction of PVB is unprofitable compared to virgin polymer, and the polymer should be dissolved in the proposed particle size.

In Figure 4, the effect of temperature on dissolution time was presented. Fixed parameters for the process were established: the stirrer speed of 500 rpm, an initial solvent volume equal to 10 mL, and polymer size 30.0 × 30.0 × 0.82 mm. It can be seen that as the temperature increases, the dissolution time decreases. At higher temperatures, solvent penetration into the polymer matrix is simplified. The mobility of the polymer segments in the solvent phase increases appreciably. PVB swells much more efficiently, and the chains disengage easily from the swollen surface. The increase in dissolution rate can be explained on a thermodynamics basis, as presented by M. T. García et al. for the polystyrene/organic solvent systems [33]. The dissolution of an amorphous polymer in a solvent is determined by the free energy of mixing: (11)$$∆G_{m}=∆H_{m}-T∆S_{m}$$
where $∆G_{m}$ is the Gibbs free energy change on mixing, $∆H_{m}$ is the enthalpy change of mixing, *T* is the temperature, and $∆S_{m}$ is the entropy change of mixing. A negative value of $∆G_{m}$ means that the mixing process will occur spontaneously. According to the Flory–Huggins theory that describes the thermodynamic equilibrium at constant pressure for polymer/solvent systems, the Gibbs free energy change of solution can be calculated as:(12)$$∆G_{m}/RT=\left(\phi_{1}/V_{1}\ln\phi_{1}+\phi_{2}/V_{2}\ln\phi_{2}+\chi_{12}\phi_{1}\phi_{2}\right)$$
where *R* is the ideal gas constant, ϕ and *V* are the volume fraction and the molar volume of components in the mixture, respectively. $\chi_{12}$ is an empirical parameter expressing the interaction enthalpy between two different molecules. The first two terms on the right side of Equation (12) denote the entropy of mixing, which quantitatively is negative since, in solution, the molecules display a more chaotic arrangement than in the solid state [34]. Moreover, the enthalpy of mixing at constant pressure can be calculated as follows: (13)$$∆H_{m}/RT=\chi_{12}\phi_{1}\phi_{2}$$

The $\chi_{12}$ parameter is always positive, therefore, $∆H_{m}$ is also positive. The greater the solubility, the lower this value is. There is a linear relationship between $\chi_{12}$ and the inverse of temperature [35]. Accordingly, when the temperature increases, the value of $\chi_{12}$ decreases, and the dissolution process is favored. Therefore, a moderate increase in temperature leads to an increase in $∆S_{m}$ and a decrease in the value of $∆H_{m}$, favoring the dissolution process and decreasing the dissolution time of polymer PVB. 

In the next step, an ultrasonic bath was used instead of a mechanical stirrer. Figure 5a shows the positive effect of ultrasound on the dissolution of PVB in IPA. The total dissolution time of the same polymer sample and amount of solvent was reduced to 4 h compared to the system where mechanical stirring and a constant temperature of 298.15 K were fixed. The following factors have an impact on this. The appropriate ultrasound power enhances the dissolution process in the liquid medium by generating and then destroying cavitation bubbles. Ultrasound is propagated via a series of compression and rarefaction waves induced in the molecules of the medium through which it passes. At sufficiently high power, the rarefaction cycle may exceed the attractive forces of the molecules of the liquid, and cavitation bubbles will form. When they collapse in succeeding compression cycles, the energy is generated for the mechanical effect [36]. The appearance of microcurrents as a result of ultrasound irradiation at the solid-liquid interface causes a significant reduction in the thickness of the diffusion layer, even in comparison to the process with intensive mixing.

The cavitation effect leads to the appearance of many microleakages on the surface of the solid subjected to ultrasound. Moreover, the elevated liquid temperature, which was caused by the ultrasound, may result in a significant effect on the dissolution rate. For this reason, the temperature of the tested system during the dissolution process was measured, as shown in Figure 5b. In the first two hours of the dissolution process, the temperature increased from ambient, *T* = 296.3 K, to *T* = 319.3 K and stabilized. In this case, the temperature stabilization is due to heat exchange between the ultrasonic bath and the environment. More effective thermal isolation of the ultrasonic bath would lead to an increase in temperature up to the solvent boiling point and a significant increase in vapor pressure in the system. However, with higher pressure operations, the cost of the industrial apparatus and the possible emission of VOCs into the atmosphere would be increased. 

In the following stage, binary solvents have been proposed to reduce the dissolution time of PVB polymer further. Based on the literature [37,38,39], it can be concluded that the use of binary solvent increases solubility due to a positive synergistic effect. Increased dissolution rates can result from an improved dissolution of plasticizers by the second solvent, improved polymer swelling, and reduced binary solution viscosity. A mixture of a good solvent, IPA, and a weak solvent or antisolvent to dissolve PVB was used to confirm the synergistic effect. A reduction in dissolution time of PVB under the same process conditions for binary mixtures: {IPA + ethyl acetate}, {IPA + butyl acetate}, {IPA + acetone}, {IPA + 2-butanone}, {IPA + acetic acid}, {IPA + toluene} in volume proportion *V*/*V* = 1/1 was obtained. A detailed analysis was carried out for the binary system {IPA + ethyl acetate, AcOEt} and presented in Figure 6, Figure 7 and Figure 8 and Appendix A in Appendix A.

As shown in Figure 6, the total polymer dissolution time was *t* = 2 h 15 min. under constant temperature, *T* = 298.15 K, and a mixing speed of 500 rpm. The PVB dissolved 4.4 times faster in the binary solution {IPA + AcOEt} than in the pure solvent, IPA. Reducing the polymer dissolution time is crucial in terms of the cost of running an industrial-scale recycling process. Similar to measurements with pure IPA, the initial volume of binary solvent does not significantly affect the time and rate of dissolution of the PVB. In the studied range of weight fraction, an almost linear correlation of dissolved polymer as a time function was observed, indicating a constant dissolution rate. Moreover, PVB dissolution was observed only up to a concentration of about 20%*wt*. Above this concentration, the viscosity of the solution increased rapidly, and the gel-like swollen layer was formed on the PVB.

The effect of temperature on the dissolution time of PVB in the binary solvent {IPA + AcOEt, *V*/*V* = 1/1} is shown in Figure 7 and Appendix A in Appendix A. Increasing the process temperature by 20 K from *T* = 298.15 K to *T* = 318.15 K significantly increased the dissolution rate of PVB and reduced the time for complete dissolution by 3.9 times.

The total dissolution time for PVB flake size 30.0 × 30.0 × 0.82 mm was *t* = 35 min. Further acceleration of the process can be obtained by shredding the polymer into smaller fractions and increasing the temperature of the process. However, the temperature process by the boiling point of the solvents is limited if the process under atmospheric pressure is carried out. At pressure, *p* = 101.3 kPa, the boiling point IPA is *T*_b_ = 355.3 K and AcOEt *T*_b_ = 350.2 K [40]. Choosing the process temperature should prevent the emission of VOCs into the atmosphere. Based on Figure 5 and Figure 7, it can be concluded that with the use of binary solvent {IPA + AcOEt} compared to pure IPA at *T* = 318.15 K, a reduction in dissolution time from *t* = 3 h 18 min to *t* = 35 min was obtained. Figure 8 shows the effect of ultrasound, where the ultrasonic bath was used instead of mechanical stirring of the sample in the PVB and binary solvent {IPA + AcOEt} system. A slightly shorter dissolution time, equal to *t* = 15 min., was obtained compared to a similar sample of PVB dissolved at *T* = 298.15 K with 500 rpm mechanical stirring. This is mainly due to the cavitation effect induced by the ultrasound wave and the temperature increase in the system during the dissolution process.

### 4.3. Density, Dynamic Viscosity, and Refractive Index of PVB Solution

From the point of view of designing and estimating the cost of the process, physicochemical properties play a key role. Furthermore, knowledge of these properties allows the obtained calibration curves to analyze polymer concentration in the solution during the dissolution process on an industrial scale. In this study, density, dynamic viscosity, and refractive index as a function of PVB concentration in a pure IPA or binary solvent {IPA + AcOEt, *V*/*V* = 1/1} were determined. Moreover, density and dynamic viscosity measurements were performed as a function of temperature and ambient pressure. The experimental data are collected in Table 2 and Table 3 and graphically presented in Figure 9 and Figure 10. The physicochemical properties of pure IPA (0%*wt* PVB) at *T* = 298.15 K with the literature data were compared in Table 2, Table 3 and Table 4, which confirms the high purity of the solvents used and the correct calibration of measuring instruments. Results show that the PVB mixture has higher density and dynamic viscosity than pure IPA or binary solvent {IPA + AcOEt}, and density and viscosity decrease with increasing temperature. In contrast to density, a temperature increase and PVB concentration decrease significantly impact the dynamic viscosity of the studied PVB mixture, inducing a sharp reduction in the viscosity. This behavior is related to thermal expansion; the liquid density and viscosity decreases, and the intermolecular interactions become weaker due to the increase in the mutual distances between the molecules [41].

The PVB polymer chain contains hydroxyl groups, which form hydrogen bonds with each other as well as with the hydroxyl group of the solvent, IPA. Hydrogen bonds break at higher temperatures, leading to lower solution viscosities. Comparing the viscosities from Figure 9b and Figure 10b for the same temperature, *T* = 298.15 K and PVB concentration, 12%*wt*, it is worth noting that the addition of the nonpolar solvent, ethyl acetate, reduces the viscosity of the solution from 1000 mPa∙s to 767 mPa∙s. AcOEt does not form hydrogen bonds; thus, intermolecular interactions in the system with binary solvents are weaker, and the viscosity is lower. From the point of view of recycling costs and reducing the emission of VOCs into the atmosphere, it is preferable to achieve the highest possible concentration of polymer solution. However, this involves a significant increase in viscosity, which will prevent further confectioning and application of the PVB solution. Therefore, the use of binary solvents, compared to IPA, is preferred to reduce the solution’s viscosity for the same polymer concentration. 

Furthermore, the density of the PVB solution increases for increasing concentrations of PVB, which can be extrapolated to the density of pure PVB; the literature density of PVB equals 1.07 g/cm^3^ [50]. It is a linear function of concentration for the measured range from 0 to 14%*wt*. PVB. The experimental data have been correlated using Equation (3) with the temperature dependence of the determined parameters using Equations (4) and (5). The value of the parameters, along with RMSE, are collected in Appendix A in Appendix A. The RMSE value between the experimental and calculated density was lower than 0.0004 g·cm^−3^ for both systems, indicating a good agreement between experimental and calculated data. 

To describe the dynamic viscosity as a function of temperature, the Andrade-type equation was used; Equation (6). Furthermore, the dependence of the equation parameters on the PVB concentration has been applied (Equations (7) and (8)). Thus, for a given temperature and concentration of the PVB solution, determining the viscosity of the solution is possible and vice versa. The calculated parameters are shown with the RMSE in Appendix A in Appendix A. A better agreement of calculated values compared to experimental data and a lower RMSE, σ = 2.5 mPa·s, was obtained for the system with pure IPA than for the system with binary solvent {IPA + AcOEt}; σ = 7.5 mPa·s. 

The refractive index as a function of PVB concentration in IPA and binary solvent {IPA + AcOEt} are depicted in Figure 9c and Figure 10c and further detailed in Table 4. The refractive index of pure PVB is 1.4850 [51] and is significantly higher than for an IPA, **n_D_ =** 1.37522 or {IPA + AcOEt} system, **n_D_** = 1.37083. Therefore, a linear refractive index increase with increasing PVB concentration was observed in the measured range. The determined linear dependence of the refractive index can be successfully used as a correlation curve to determine the concentration of PVB polymer in the test solution.

### 4.4. UV−VIS Spectrum for Liquid PVB Solution

Figure 11 exhibits the UV-VIS absorption spectra of 10%*wt* PVB solution in IPA and {IPA + AcOEt, *V*/*V* = 1/1} at room temperature. Both spectra are similar, showing full transmittance of radiation in the visible range and absorption of UV radiation. The obtained spectrum is comparable to the ones PVB/ethanol solution presented in the literature [52]. The absorption bands observed in the UV range are a result of the hydroxyl groups present in the polymer chain. The n to π* electron transitions are characteristic of the lone pair of oxygen electrons of the OH group or the cyclic acetyl electrons C=O and C-O-C present in the polymer chain; all functional groups involved in intramolecular and intermolecular hydrogen bonding [53]. From an application point of view, the PVB solution can be used as a clear varnish, which will form a transparent film once the solvent has evaporated. Moreover, the PVB protective coating provides a barrier to UV rays, which is a major advantage.

## 5. Conclusions

The present study investigated the effects of temperature, polymer size, and solvent volume on the dissolution time of polyvinyl butyral used in laminated glass. The following solvents were used: 2-propanol and a binary mixture of 2-propanol and ethyl acetate. The results were as follows: The shortest dissolution time of PVB, *t* = 35 min for the system with {2-propanol + ethyl acetate, *V*/*V* = 1/1} at *T* = 318.15 K, the stirrer speed of 500 rpm, and PVB size 30.0 × 30.0 × 0.82 mm was obtained. The reduced dissolution time of PVB is determined by polymer fragmentation and higher temperatures, *T* = 318.15 K. PVB dissolves much faster in binary solvents than in pure alcohol under the same conditions. Moreover, a positive effect on reducing dissolution time using ultrasound was observed. Experimental and calculated data of the density, dynamic viscosity, and refractive index of PVB liquid solutions as a function of concentration and temperature can be used to design a system of filtration. Complete removal of solid contaminants, especially heating wires and glass dust, will only be possible after polymer dissolution, filtration or decantation. The results obtained are of great importance in the design of a PVB recycling process line, as well as in the preparation of PVB polymer-based solutions that can be used as impregnates, varnishes, and structural adhesives for bonding glass, metal, wood, paper, leather, cloth, and other materials. In subsequent studies, other binary mixtures, e.g., alcohol with acetone, 2-butanone, butyl acetate, acetic acid, toluene, or xylene, should be the research aim. The appropriate solvents for the above-mentioned applications should be selected. Moreover, using these binary mixtures can reduce the dissolution time of the PVB and design the physicochemical parameters of the PVB solution: a lower dynamic viscosity with a higher polymer concentration and different evaporation rates, which is important, e.g., the adhesive cure time and impregnation.

## Figures and Tables

**Figure 1 polymers-14-05119-f001:**
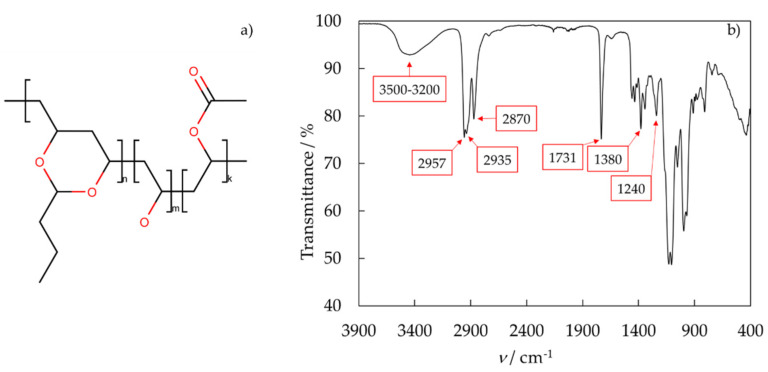
(**a**) Structure and (**b**) FT″−IR-spectra of the PVB.

**Figure 2 polymers-14-05119-f002:**
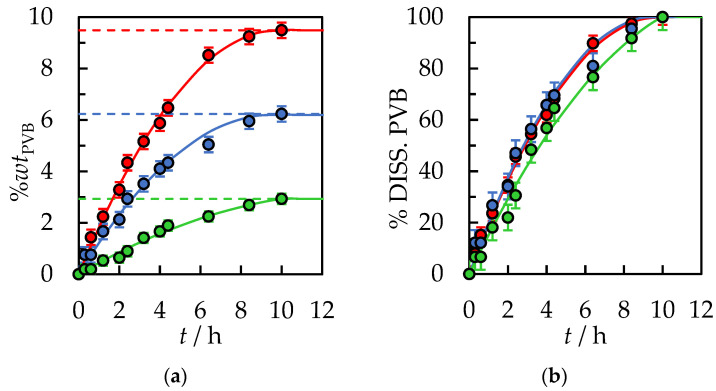
(**a**) %*wt* of PVB in IPA and (**b**) % of dissolved PVB as a function of the dissolution time at *T* = 298.15 K, the stirrer speed of 500 rpm, and polymer size 30.0 × 30.0 × 0.82 mm. Experimental data: ● for 30 mL, ● for 15 mL, ● for 10 mL of initial solvent volume. Solid lines are correlation data using Equation (1).

**Figure 3 polymers-14-05119-f003:**
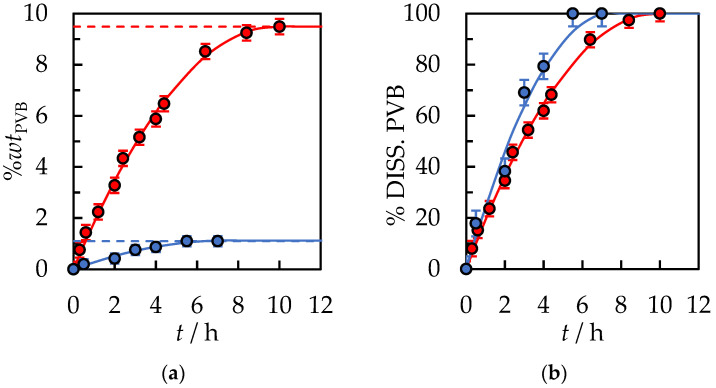
(**a**) %*wt* of PVB in IPA and (**b**) % of dissolved PVB as a function of the dissolution time at *T* = 298.15 K, the stirrer speed of 500 rpm, and initial solvent volume, *V*_IPA_ = 10 mL. Polymer size: ●—30.0 × 30.0 × 0.82 mm, ●—10.0 × 10.0 × 0.82 mm. Solid lines are correlation data using Equation (1).

**Figure 4 polymers-14-05119-f004:**
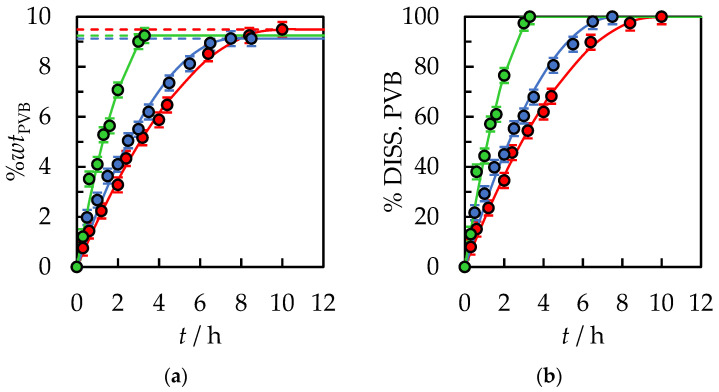
(**a**) %*wt* of PVB in IPA and (**b**) % of dissolved PVB as a function of the dissolution time. The stirrer speed of 500 rpm, initial solvent volume, *V*_IPA_ = 10 mL, and polymer size 30.0 × 30.0 × 0.82 mm. Temperature: ●—298.15 K, ●—308.15 K, ●—318.15 K. Solid lines are correlation data using Equation (1).

**Figure 5 polymers-14-05119-f005:**
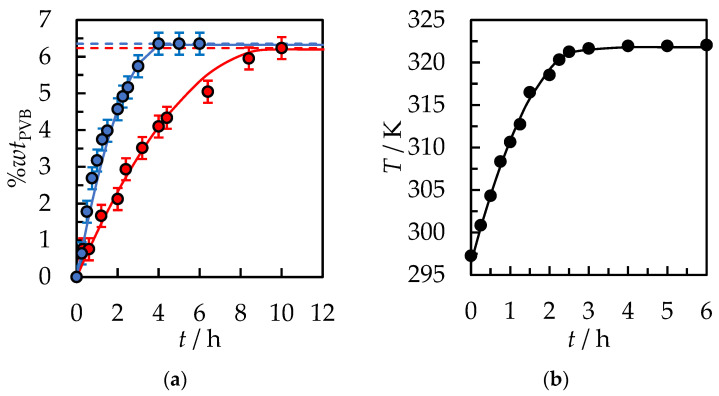
(**a**) %*wt* of PVB in IPA as a function of the dissolution time. Initial solvent volume, *V*_IPA_ = 15 mL, polymer size 30.0 mm × 30.0 mm × 0.82 mm. ●—*T* = 298.15 K, The stirrer speed of 500 rpm. ●—ultrasonic mixing. Solid lines are correlation data using Equation (1) (**b**) Temperature of the tested system during the dissolution process. A solid line is a guide for the eye.

**Figure 6 polymers-14-05119-f006:**
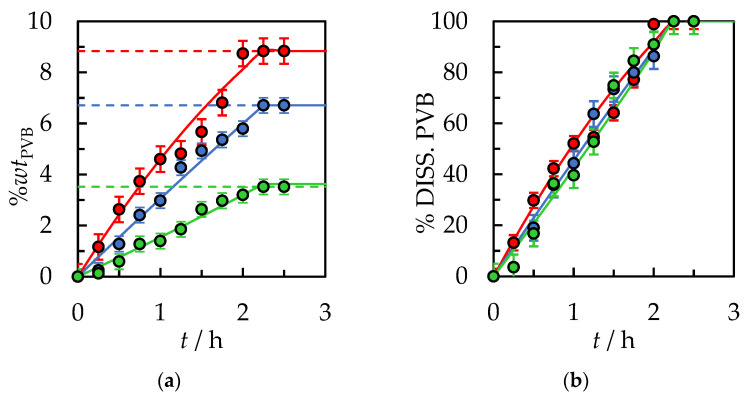
(**a**) %*wt* of PVB in {IPA + AcOEt, *V*/*V* = 1/1} and (**b**) % of dissolved PVB as a function of the dissolution time at *T* = 298.15 K, the stirrer speed of 500 rpm, and polymer size 30.0 × 30.0 × 0.82 mm. Experimental data: ● for 30 mL, ● for 15 mL, ● for 10 mL of initial solvent volume. Solid lines are correlation data using Equation (1).

**Figure 7 polymers-14-05119-f007:**
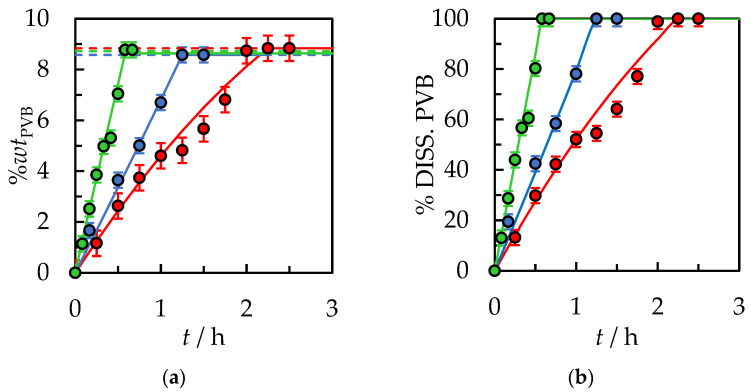
(**a**) %*wt.* of PVB in {IPA + AcOEt, *V*/*V* = 1/1} and (**b**) % of dissolved PVB as a function of the dissolution time. The stirrer speed of 500 rpm, initial solvent volume, *V*_IPA_ = 10 mL, and polymer size 30.0 × 30.0 × 0.82 mm. Temperature: ●—298.15 K, ●—308.15 K, ●—318.15 K. Solid lines are correlation data using Equation (1).

**Figure 8 polymers-14-05119-f008:**
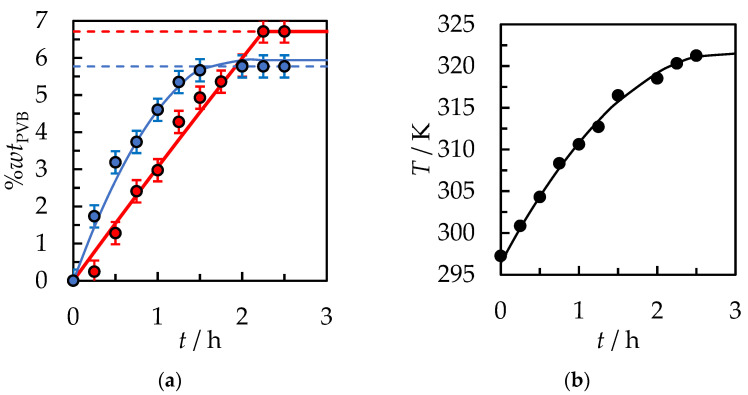
(**a**) %*wt* of PVB in {IPA + AcOEt, *V*/*V* = 1/1} as a function of the dissolution time. Initial solvent volume, *V*_IPA_ = 15 mL, polymer size 30.0 × 30.0 × 0.82 mm. ●—*T* = 298.15 K, The stirrer speed of 500 rpm. ●—ultrasonic mixing. Solid lines are correlation data using Equation (1) (**b**) Temperature of the ultrasonic bath. A solid line is a guide for the eye.

**Figure 9 polymers-14-05119-f009:**
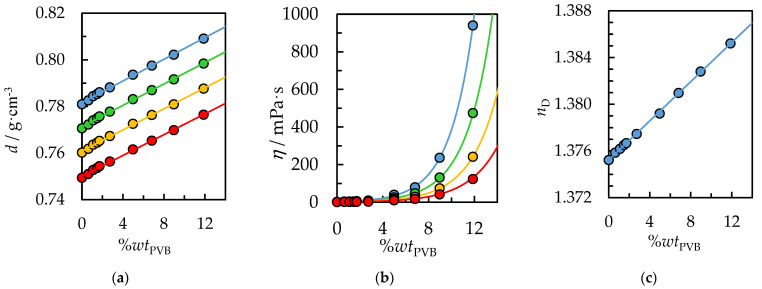
(**a**) Density, *d*, (**b**) dynamic viscosity, *η*, and (**c**) refractive index, *n*_D_, as a function of *wt.* % PVB in IPA and temperature. Experimental data: ● for *T* = 298.15 K, ● for *T* = 308.15 K, ● for *T* = 318.15 K, ● for *T* = 318.15 K. Solid lines are correlation data using Equation (3) for density, Equation (6) for dynamic viscosity and Equation (9) for the refractive index.

**Figure 10 polymers-14-05119-f010:**
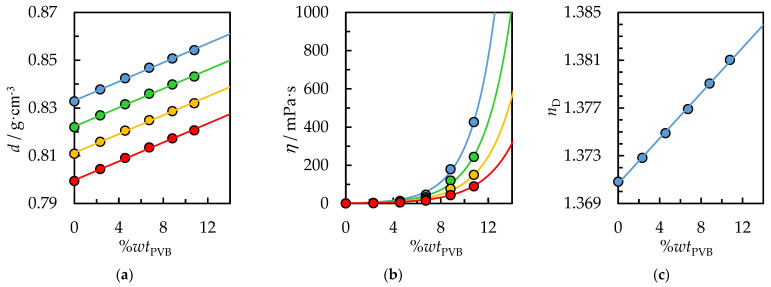
(**a**) Density, *d*, (**b**) dynamic viscosity, *η,* and (**c**) refractive index, *n*_D,_ as a function of %*wt* PVB in {IPA + AcOEt, *V*/*V* = 1/1} and temperature. Experimental data: ● for *T* = 298.15 K, ● for *T* = 308.15 K, ● for *T* = 318.15 K, ● for *T* = 318.15 K. Solid lines are correlation data using Equation (3) for density, Equation (6) for dynamic viscosity and Equation (9) for the refractive index.

**Figure 11 polymers-14-05119-f011:**
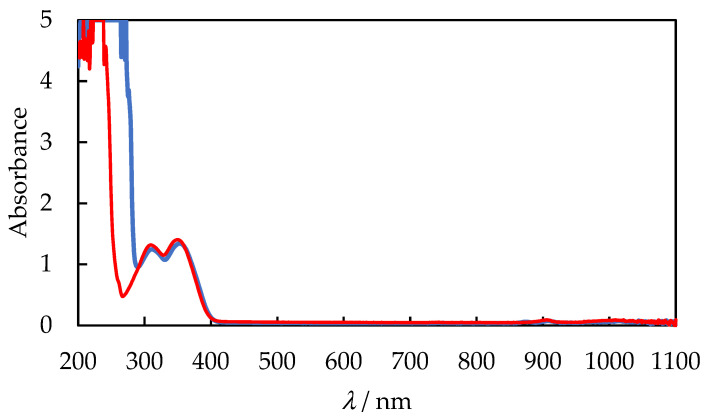
UV−VIS absorption spectra for 10%*wt* PVB liquid solution in IPA—blue line and {IPA + AcOEt, *V*/*V* = 1/1}—red line.

**Table 1 polymers-14-05119-t001:** Specification of chemical components.

Chemical Name	Abbreviation	CAS No.	Supplier	Mass Fraction Purity	Purification Method
Polyvinyl (butyral)	PVB	63148-65-2	Pilkington Automotive	-	None
2-Propanol	IPA	67-63-0	Avantor	>0.99 ^1^	None
Ethyl acetate	AcOEt	141-78-6	Chempur	>0.99 ^1^	None
Acetone	Ace	67-64-1	Chempur	>0.99 ^1^	None

^1^ Stated by the supplier.

**Table 2 polymers-14-05119-t002:** Density, *d*, of PVB solution in 2-propanol, or binary mixture {2-propanol + ethyl acetate *V*/*V* = 1/1} at *p* = 100 kPa.

	*d*/g·cm^−3^			
%*wt* PVB	*T* = 298.15 K	*T* = 308.15 K	*T* = 318.15 K	*T* = 328.15 K
PVB + 2-propanol				
0	0.7809 0.78089 [42] * 0.78088 [43] * 0.78098 [44] *	0.7706	0.7602	0.7494
0.63	0.7826	0.7723	0.7618	0.7510
1.10	0.7844	0.7740	0.7636	0.7528
1.49	0.7853	0.7749	0.7645	0.7536
1.72	0.7861	0.7757	0.7653	0.7544
2.73	0.7882	0.7778	0.7673	0.7564
4.98	0.7936	0.7831	0.7726	0.7616
6.82	0.7975	0.7870	0.7763	0.7653
8.96	0.8022	0.7916	0.7809	0.7698
11.88	0.8091	0.7984	0.7876	0.7765
PVB + {2-propanol + ethyl acetate *V*/*V* = 1/1}				
0	0.8328	0.8220	0.8109	0.7994
2.33	0.8378	0.8269	0.8159	0.8044
4.57	0.8424	0.8316	0.8205	0.8091
6.74	0.8469	0.8360	0.8249	0.8135
8.82	0.8507	0.8399	0.8287	0.8173
10.79	0.8542	0.8432	0.8320	0.8206

%*wt* PVB is the weight percent of PVB in solution. Standard uncertainties are *u*(*T*) = 0.05 K, *u*(*p*) = 2 kPa, *u*(%*wt PVB*) = 0.05, *u*(*d*) = 5·10^−4^ g·cm^−3^. * The literature value for pure IPA.

**Table 3 polymers-14-05119-t003:** Dynamic viscosity, *η,* of PVB solution in 2-propanol, or binary mixture {2-propanol + ethyl acetate *V*/*V* = 1/1} at *p* = 100 kPa.

	*η*/mPa·s			
%*wt* PVB	*T* = 298.15 K	*T* = 308.15 K	*T* = 318.15 K	*T* = 328.15 K
PVB + 2-propanol				
0	2.06 2.089 [43] * 2.086 [45] * 2.045 [46] *	1.54	1.18	0.92
0.63	3.98	2.94	2.20	1.64
1.10	4.16	3.03	2.23	1.64
1.49	4.79	3.44	2.50	1.81
1.72	6.05	4.32	3.11	2.24
2.73	9.61	6.62	4.61	3.20
4.98	39.9	25.5	16.4	10.6
6.82	79.4	47.6	28.8	17.5
8.96	237	132	74.2	41.8
11.88	940	474	242	123
PVB + {2-propanol + ethyl acetate *V*/*V* = 1/1}				
0	0.909	0.789	0.693	0.639
2.33	3.30	2.66	2.15	1.72
4.57	13.5	10.1	7.64	5.27
6.74	45.8	32.5	22.9	15.5
8.82	179	120	77.4	44.0
10.79	426	244	150	89.9

%*wt* PVB is the weight percent of PVB in solution. Standard uncertainties are *u*(*T*) = 0.05 K, *u*(*p*) = 2 kPa, *u*(%*wt PVB*) = 0.05, *u*(*η*) = 5%. * The literature value for pure IPA.

**Table 4 polymers-14-05119-t004:** Refractive index, *n*_D_, of PVB solution in 2-propanol or binary mixture {2-propanol + ethyl acetate *V*/*V* = 1/1} at *T* = 298.15 K and *p* = 100 kPa.

%*wt* PVB	*n* _D_	%*wt* PVB	*n* _D_
PVB + 2-propanol		PVB + {2-propanol + ethyl acetate *V*/*V* = 1/1}	
0	1.37522 1.37515 [47] * 1.3752 [48] * 1.3749 [49] *	0	1.37083
0.63	1.37585	2.33	1.37282
1.10	1.37618	4.57	1.37489
1.49	1.37651	6.74	1.37691
1.72	1.37667	8.82	1.37905
2.73	1.37746	10.79	1.38103
4.98	1.37921		
6.82	1.38097		
8.96	1.38280		
11.88	1.38521		

%*wt* PVB is the weight percent of PVB in solution. Standard uncertainties are *u*(*T*) = 0.05 K, *u*(*p*) = 2 kPa, *u*(%*wt PVB*) = 0.05, *u*(*n*_D_) = 5·10^−5^. * The literature value for pure IPA.

## Data Availability

The data presented in this study are available in the article and Appendix A.

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
