# Peer review of "Selection of Conditions in PVB Polymer Dissolution Process for Laminated Glass Recycling Applications"

_polymers, 2022, doi:10.3390/polym14235119_

Round 1

Reviewer 1 Report

The manuscript by these Authors deals with the setting of the best operating parameters to provide the Polyvinyl(butyral) based polymeric waste. The study is interesting and the results are worthy of publication. Anyway, I would point out that the text needs a revision to improve language and scientific aspects (the experimental section must be more detailed). I just want to highlight the request to the Authors of checking the correct DSC calibration temperature range, Indium melting temperature is 430 K, in order to proceed with calibration you need to close the melting stage, thus you need to finish the scan at least at 440 K. How is possible you closed at 430 K? My remarks and suggestions are reported in the attached pdf.

Author Response

Response to review:

I would suggest to strenghen this statement by citing some relevant researches as regards the polymers' treatment at their end of life, such as: Procedia Manufacturing Volume 8, 2017, Pages 649-656 Procedia Manufacturing Drivers to Sustainable Plastic Solid Waste Recycling: A Review Lifetime prediction of food and beverage packaging wastes. J Therm Anal Calorim 125, 809–816 (2016). https://doi.org/10.1007/s10973-015-5169-9

Thank you for your comment. The appropriate reference has been added.

The operating parameters are not reported, atmosphere, scanning rate, temperature range of investigation, number of replicates, average sample mass used.

We agree with the reviewer. Our oversight. Operating parameters have been added.

How is possible? Indium melting temperature is 430, in order to proceed with calibration you need to close the melting stage, thus you need to finish the scan at least at 440 K.

Yes, we agree with the reviewer. The melting point of indium is T = 429.8 K under p = 1 atm. We have given the temperature range for which the calibration is valid. However, the measurement for indium was performed over a wider range, up to 450 K. This has been corrected.

Minor editorial errors have been corrected according to the reviewer's instructions.

Reviewer 2 Report

The authors have performed extensive experiments in PVB polymer dissolution process for laminated glass recycling applications. Manuscript can be accepted after a minor revision.

·         Briefly discuss the environmental issue and cost for the PVB polymer-based solutions.

·         In the methodology section, it is recommended to add some images of the testing samples, process, etc.

·         For easy understanding by the readers, it will be nice to add the acceptable range of different parameters of the solutions such as transition temperature, density, dynamic viscosity, refractive index, etc.

·         Conclusion should be rewritten by highlighting the major findings quantitatively in bullet points. Also, limitations of the study together with future research scope must discuss briefly.

Author Response

The authors have performed extensive experiments in PVB polymer dissolution process for laminated glass recycling applications. Manuscript can be accepted after a minor revision.

Briefly discuss the environmental issue and cost for the PVB polymer-based solutions.

The relevant paragraph has been added in the introduction.

In the methodology section, it is recommended to add some images of the testing samples, process, etc.

The images of the studied PVB have been added to the Supplementary Materials. In my opinion they are not that important to be in the main text.

For easy understanding by the readers, it will be nice to add the acceptable range of different parameters of the solutions such as transition temperature, density, dynamic viscosity, refractive index, etc.

The ranges of the various physicochemical parameters have been added in the experimental section.

Conclusion should be rewritten by highlighting the major findings quantitatively in bullet points. Also, limitations of the study together with future research scope must discuss briefly.

The conclusions have been revised according to the reviewer's recommendations.